# Grammatical impairment in schizophrenia: An exploratory study of the pronominal and sentential domains

**Monica F. Chaves** [1]*, **Cilene Rodrigues**[1], **Sidarta Ribeiro**[2], **Natália B. Mota**[3], **Mauro Copelli**[4]

**1** Department of Letters and Literature, Pontifical Catholic University of Rio de Janeiro, PUC-Rio, Rio de Janeiro, RJ, Brazil, **2** Brain Institute, Federal University of Rio Grande do Norte, UFRN, Natal, RN, Brazil, **3** Department of Psychiatry, Federal University of Rio de Janeiro, UFRJ, Rio de Janeiro, RJ, Brazil, **4** Department of Physics, Federal University of Pernambuco, UFPE, Recife, PE, Brazil

* monicachaves64@esp.puc-rio.br

**Data Availability Statement:** All relevant data are within the paper and its Supporting Information files.

## Abstract

Schizophrenia (SZ) is a severe mental disorder associated with a variety of linguistic deficits, and recently it has been suggested that these deficits are caused by an underlying impairment in the ability to build complex syntactic structures and complex semantic relations. Aiming at contributing to determining the specific linguistic profile of SZ, we investigated the usage of pronominal subjects and sentence types in two corpora of oral dream and waking reports produced by speakers with SZ and participants without SZ (NSZ), both native speakers of Brazilian Portuguese. Narratives of 40 adult participants (20 SZ, and 20 NSZ–sample 1), and narratives of 31 teenage participants (11 SZ undergoing first psychotic episode, and 20 NSZ–sample 2) were annotated and statistically analyzed. Overall, narratives of speakers with SZ presented significantly higher rates of matrix sentences, null pronouns—particularly null 3Person referential pronouns—and lower rates of non-anomalous truncated sentences. The high rate of matrix sentences correlated significantly with the total PANSS scores, suggesting an association between the overuse of simple sentences and SZ symptoms in general. In contrast, the high rate of null pronouns correlated significantly with positive PANSS scores, suggesting an association between the overuse of null pronominal forms and the positive symptoms of SZ. Finally, a cross-group analysis between samples 1 and 2 indicated a higher degree of grammatical impairment in speakers with multiple psychotic episodes. Altogether, the results strengthen the notion that deficits at the pronominal and sentential levels constitute a cross-cultural linguistic marker of SZ.

## Introduction

Schizophrenia (SZ) is a condition of the mind/brain characterized by a loss of the ability to form coherent and complex ideas about oneself and the world [1]. Disturbances of thought and language are hallmark features of SZ [2–7]; however, only language can be directly observed. Thus, language disturbances are considered fundamental tools for the diagnostic of

**Funding:** MCF: Research fellowships from the Brazilian Coordination for the Improvement of Higher Education Personal (Capes) and the Brazilian National Council for Scientific and Technological Development (CNPq) CR: The Brazilian National Council for Scientific and Technological Development (CNPq) Humanities. Grant number: 439434/ 2018-1. The funders had no role in study design, data collection and analysis, decision to publish, or preparation of the manuscript.

**Competing interests:** The authors have declared that no competing interests exist.

SZ, with many diagnostic criteria being based on the patient's abnormal language production (e.g., clang associations, alogia, looseness of associations, poverty of speech, derailment) [8].

The centrality of the language cognitive system in SZ is emphasized by Tim Crow [9–12], whose evolutionary theory explores the idea that SZ is a species related disease, characterized by language disturbances due to specific alterations on the neuronal system that connect the cerebral hemispheres [13, 14]. Based on Crow's view of SZ, but with "linguistic lens", Hinzen & Rosselló [15] argue that the core symptoms of SZ–delusions (i.e., false or bizarre utterances), auditory verbal hallucinations (AVH) (i.e., hearing voices inside your head) and formal thought disorder (FTD) (i.e., disorganized speech)–are not only language related, but also indicative of a breakdown of the language cognitive system *per se.* The general idea is that Grammar in the face of SZ has a specific linguistic profile, which differs from neurotypicals' profile. Moreover, the link between SZ and language dysfunctions is supported by current findings indicating associations between FTD and structural and functional anomalies in language areas of the brain [16, 17], as well as an overlap between FTD and AVH, even in early stages of SZ, with substantial shared anomalies in language areas of the brain, and gene overlap between language dysfunctions and SZ [18, 19].

FTD, although traditionally conceptualized as a disruption of normal flow of thought, is in fact a complex syndrome diagnosed by the patient's speech behavior. FTD comprises a set of positive (or psychotic) symptoms–characterized by disorganized discourse (e.g., tangentiality, derailment, illogicality and incoherence)–, and set of negative symptoms–characterized by reduction in speech production and impoverishment of speech content (e.g., alogia, poverty of speech, poverty of content of speech)–, which are associated with the SZ even during first episode [20–22].

Albeit FTD symptoms are commonly assessed by psychometric rating scales, which relies on third party's observations, recently, more direct approaches to language anomalies have been developed. In this arena of research, studies adopting speech samples (e.g., social media, written texts, interviews, narratives etc.) have already shown great potential in identifying syntactic (e.g., reduced syntactic complexity, referential failures, misuse of pronouns) and semantic (e.g., word similarity) features of the linguistic profile of SZ. However, there is still much to be done before we can identify a language-specific profile of SZ. Our contribution, by examining language anomalies in SZ from a formal grammatical view of language, is to provide a more fine-grained investigation on how, why and where Grammar is hinged in SZ, and while doing so, verifying whether crosslinguistic differences affect the observed deficits.

Notwithstanding the scarcity of studies adopting a formal grammatical approach to the investigation of language in SZ, there is much evidence that the disorder at hand affects the form and content of language. Little semantic connectivity, overuse of pronouns which lack a clear referent, and preference for indefinite nominal expressions linked to difficulties in contextualizing definite noun phrases indicate impairments within the semantic and pragmatic components of language [23–30]. The overproduction of shorter and simpler sentences, with avoidance of sentential embedding, the high amount of morphosyntactic errors related to verb agreement and tense morphology and the frequent violations of syntactic constraints all point towards major deficits in syntax and morphology [31–37].

Studies focusing on nominal expressions have shown that the speech and writing of SZ patients, when compared to controls, present an overuse of either first person singular or third person pronouns [38–43], and higher proportions of referential anomalies in the usage of definite nominal expressions [29, 30, 44]. At the sentence level, speakers diagnosed with SZ show a notable reduction of syntactic structure, producing fewer embedded sentences [30, 36, 45], with a greater occurrence of ungrammatical truncation (ungrammatical incomplete sentences) within embedded clauses [46]. Also, comprehension studies indicate that participants with SZ

are less sensitive to syntactic constraints, such as those underlying question formation [37]. Comprehension studies also indicate that participants with SZ present deficits in deriving the meaning of embedded clauses [4, 47, 48]. Altogether, these studies support the hypothesis that SZ is a disorder linked to language [7, 12, 15, 49] that, by affecting grammatical knowledge, may impair one's ability to build complex grammatical structures [15].

Although the brain networks responsible for language are essentially the same crosslinguistically [50], languages differ from each other with respect to grammatical choices. Which means that, in order to characterize the language-specific profile of SZ, it is important to take into consideration the idea that different types of neurocognition might correlate with different linguistic types. For example, while English does not license null pronouns, Romance languages like Spanish and Italian readily do in subject position. Hence, given that SZ is universal, being found within different linguistic communities, outlining its linguistic profile requires a comprehensive understanding of grammatical variation across languages. Tovar et al.'s study on the usage of pronouns in a sample of Spanish speakers with SZ shows a significantly higher rate of referential anomaly with null pronouns as compared to overt pronouns [51]. This is not observable in a non-null-subject language, such as English.

To further the investigation of referential and syntactic deficits in SZ, we conducted an exploratory study of the grammar of pronouns and sentences in narratives produced by native speakers of Colloquial Brazilian Portuguese (CBP).

CPB is a partial null subject language. Partial null subject languages, in contrast with full null subject languages such as Spanish, European Portuguese and Italian, impose syntactic and semantic constraints on 3Person null referential pronouns, restricting their occurrence to embedded clauses and forcing them to take the structurally closest noun phrase as their referential antecedent [52–55]. To exemplify this, consider the sentences in (1) and (2). In European Portuguese, sentence (1) is grammatical, with the null subject (represented by ø) referring to the topic of the conversation (let's say *Pedro*). Likewise, in (2), the embedded null subject can refer either to *Pedro* or to *João*. Hence in full null subject languages, null pronouns behave semantically and syntactically in the same way as overt pronouns. By contrast, in CBP, null subject pronouns behave like anaphors; they are not licensed in matrix clauses and, when they occur within embedded clauses, they refer back to the closest noun phrase. Thus, sentence (1) is ungrammatical in CBP, and sentence (2) is grammatical only if the embedded null subject refers back to *João*.

(1) ø está doente

> is sick
> 'He is sick.'

(2) Pedro disse que João pensa que ø está doente

> Pedro said that João thinks that is sick
> 'Pedro said that João thinks that he is sick.'

In addition, the rate of null subject pronouns in subject position is low in CBP. Comparative investigations between CBP and European Portuguese indicate that only 29% of referential pronouns in subject position are null in CBP, whereas in European Portuguese the rate is 67% [56–58]. This low rate is also observed in language acquisition [59]. Therefore, CBP provides very interesting empirical grounds to test the usage of null pronouns in SZ.

In the present study we set out to determine whether CBP speakers with SZ are particularly sensitive to the semantic and grammatical restrictions imposed upon null subject pronouns. We set our exploratory investigation of this issue by examining two corpora of oral narratives consisting of dream and waking reports produced by SZ and NSZ native speakers of CBP.

These corpora were previously collected for the purpose of graph-theoretical studies of SZ [28, 60], which have shown that speech graph attributes obtained from dream narratives (DM), as opposed to waking narratives (WK), could differentiate between participants with SZ, participants with bipolar disorder type I, and NSZ participants [60]. Furthermore, the speech graph attributes of DM and negative image reports produced by teenaged speakers undergoing psychosis could discriminate between the three groups of speakers, with dream reports showing the best classification [28]. We conducted a syntactic analysis of these corpora focusing on (a) 3Person referential null and overt pronominal forms in subject position, which were further screened for reference anomaly, and (b) sentence complexity, considering matrix and embedded structures. Sentential truncation was also computed to verify speakers' ability to produce grammatical ellipsis at the sentential level. Based on the results of the current literature on SZ and language, we worked with the following predictions with respect to the SZ groups:

P.1: Overuse of null pronominal forms.

P.2: Higher rate of referentially anomalous 3Person pronouns.

P.3: Overuse of simple sentential structures, displaying more matrix and fewer embedded sentences.

P.4: Difficulties with sentential truncation, exhibiting a higher rate of ungrammatical ellipsis.

## Materials and methods

### Corpora

The two corpora of narratives examined in the present study were previously collected for the purpose of graph-theoretical studies [28, 60]. Dream reports (DM) were elicited by the prompting request "please report a recent dream" and waking reports (WK) by the prompting request "please report your waking activities immediately before that dream". The corpora of interviews examined in our first sample consisted of a subset of Mota's et al. (2014) and, in our second sample, a subset of Mota's et al. (2017), considering DM and WK narratives of the SZ and NSZ groups only. The group of participants with SZ of sample 1 consisted of adults with chronic SZ, and the group participants with SZ of sample 2 consisted of adolescents with first episode SZ. Our choice for keeping two sample groups was based on studies showing structural anomalies in language areas of the brain in patients with chronic SZ [61], and language differences between patients with chronic and first episode SZ [34]. The study in its totality was presented and approved by the PUC-Rio (Pontifical Catholic University of Rio de Janeiro) Research Ethics Committee (permit 26.2019).

### Participants and narrative samples

**Sample 1.** Twenty DSM-IV patients with SZ, who were all native speakers of Brazilian Portuguese, were recruited in a local mental hospital and in a mental clinic within the area of Natal/Rio Grande do Norte, Brazil (**Table 1**). All patients were on antipsychotic medication at the time of the assessment, three on a mood stabilizer, and five on benzodiazepine. Twenty NSZ participants were recruited in their workplace in Natal/Rio Grande do Norte, Brazil (**Table 1**). All NSZ participants were neurotypical individuals, five with depression, two with generalized anxiety disorder, one with one past episode of post-traumatic stress disorder, and eleven with various symptoms of mood/anxiety disorder without reaching diagnostic criteria, plus one healthy individual. At the time of the assessment, three were taking benzodiazepine and four were taking antidepressants. Participants of both groups were independently

**Table 1. Features of SZ and NSZ groups of Sample 1 (adapted from Mota et al. 2014).**

|  | Participants with SZ (*n* = 20) | NSZ participants (*n* = 20) | *P* values (SZ x NSZ) |
|---|---|---|---|
| *Age (years)* | 34.79 ± 9.60 | 35.05 ± 11.21 | 0.978 |
| *Sex (males/females)* | 16/4 | 9/11 | 0.022 |
| *Years of Education* | 6.85 ± 4.37 | 9.80 ± 4.35 | 0.052 |
| *PANSS* [a] | 69 ± 16.59 | 36.15 ± 6.43 | < 0.001 |
| *BPRS* [b] | 16.40 ± 7.26 | 3.95 ± 3.72 | < 0.001 |
| *Medication (yes/no)* | 20/0 | 7/13 | < 0.001 |
| *Age of Onset* | 22.5 ± 7.67 | 36.8 ± 7.96 | |
| *Disease Duration (years)* | 12 ± 8.3 | 1.24 ± 1.4 | |

[a] PANSS = Positive and Negative Syndrome Scale.

[b] BPRS = Brief Psychiatric Rating Scale.

evaluated by the standard Structured Clinical Interview for DSM-IV rating SCID (Portuguese version) and examined for major changes in state and level of consciousness (e.g., drowsiness), signs of autopsychic and allopsychic disorientation (e.g., inability to remember name, age, spatial localization), and reduced mnemonic and cognitive capacity. During the interviews, the "Positive and Negative Syndrome Scale" (PANSS) and the "Brief Psychiatric Rating Scale" (BPRS) were applied to all participants to quantify symptoms, including psychosis. Exclusion criteria were based on the psychosis being caused by neurological disorder or induced by psychotropic substances, on lack of any dream memory, and on refusal to participate in the experiment. All participants were voluntarily recruited and signed a consent form.

The sample of narratives of sample 1 consists of the interviews of the SZ and NSZ groups of Mota et al. (2014): 40 DM reports (20 SZ and 20 NSZ), and 40 WK reports (20 SZ group and 20 NSZ).

**Sample 2.** Eleven teenage DSM-IV patients with SZ were recruited in a local public child psychiatric clinic within the area of Natal/Rio Grande do Norte, Brazil. They were interviewed and psychometrically evaluated on the first clinical contact for recent-onset psychosis, with the disorder and diagnosis being established six months after the interview. Of the SZ group, 55% of the participants were under typical antipsychotic medication and 82% under atypical antipsychotic medication, 9% were taking mood stabilizers, 9% were taking benzodiazepine, and 9% were taking antidepressants. Exclusion criteria were based on the psychosis being drug-related or it being caused by neurological disorders. Twenty NSZ neurotypical participants were recruited in local schools around Natal, Rio Grande do Norte, Brazil. Sample 2 NSZ participants were interviewed, but not psychometrically evaluated. Exclusion criteria were based on the participants having any psychiatric symptom or diagnosis, as informed by family members during the interview.

All participants were voluntarily recruited, signed a consent form, and then asked to report a recent dream and an account of the events of the day before the dream. All reports were limited to 30 sec. Our final sample consisted of DM and WK produced by 31 participants (11 SZ and 20 NSZ). Socio-demographic information of all 31 participants and the psychiatric information of the group of participants with SZ are given in **Table 2**.

**Clinical significance of the sample.** The samples' sizes were defined in Mota et al. (2014 and 2017) and are based on world and Brazil's prevalence of schizophrenia [62]. Estimation of adequate sample size (N) considered the following equation: $n = \frac{Z^2 P(1-P)}{d^2}$

Where *Z* stands for the statistic for a level of confidence, *P* stands for expected prevalence or proportion, and *d* stands for precision. We adopted a conventional level of confidence

**Table 2. Features of SZ and NSZ groups of Sample 2 (adapted from Mota et al. 2017).**

|  | Participants with SZ (*n* = 11) | NSZ participants (*n* = 20) | *P* values (SZ x NSZ) |
|---|---|---|---|
| *Age (years)* | 14.64 ± 2.69 | 15.80 ± 3.30 | 0.476 |
| *Sex (males/females)* | 9/2 | 11/9 | 0.135 |
| *Years of Education* | 5.73 ± 2.45 | 8.35 ± 2.54 | 0.016 |
| *PANSS* [a] | 69.27 ± 13.91 |  |  |
| *BPRS* [b] | 16.73 ± 5.88 |  |  |
| *Disease Duration (days)* | 339.36 ± 244.80 |  |  |

[a] PANSS = Positive and Negative Syndrome Scale.

[b] BPRS = Brief Psychiatric Rating Scale.

interval of 95%, with $Z = 1.96$, and $d = 0,05$. A review of 46 countries data, consisting of 154,140 cases, considered the lifetime prevalence of schizophrenia to be 0.55% (±0.45 SD). Whereas studies focused on the Brazilian population report a prevalence of 0.57% for schizophrenia [63]. The estimate sample size of N = 9 is reached based on either expected prevalence (0,57% and 0,55%). Importantly, no estimated sample size was greater than N = 20, with N < 10 for mean lifetime prevalences in the world sample.

## Annotation scheme

Prior to annotating the corpora of narratives, the available audios and the original transcripts of Mota et al. (2014 and 2017) were checked against each other. The final text of each narrative was divided into units of speech corresponding to sentences (i.e., units containing a subject and a predicate and conveying a proposition). All sentences with finite verbs were identified. Full sentences were coded either as matrix or embedded, syntactically incomplete sentences were tagged as truncated and as non-anomalous or anomalous, depending on the recoverability of their contents and grammaticality. Agreement errors were not annotated given that agreement is subject to dialectal variation in CBP. Subject pronouns occurring in the context of finite predicates were tagged as null, for those with no phonological content, and as overt, for those with phonological content. Null 3Person subject pronouns were coded as referential for pronouns with a referential interpretation, and non-referential for expletives and for all 3Person null pronouns with impersonal and generic readings. Referential 3Person pronouns, both overt and null, were tagged as anomalous if their referents were missing, unclear, or ambiguous, and non-anomalous if their referents were clear.

All transcripts were coded by one annotator (first author), and 37% of the data (15 transcripts in sample 1, and 12 transcripts in sample 2) were randomly selected and distributed to two raters (MC and CP), doctors in theoretical linguistics (syntax), not involved in the present research, being, thus, blind to group status. All annotations were later checked by a senior rater, also blind to group status (second author). Under the supervision of the senior rater, all annotated transcripts were later compared, and disagreements were discussed before a final decision was reached. While the transcripts of the narratives of Mota et al. (2017) included all interviewer-participant interactions, the interviewers' comments and questions were missing in some of the narrative transcripts of Mota et al. (2014). Thus, of the final samples of the 40 DM analyzed in sample 1, 20 were missing the interviewers' questions (9 of the NSZ group and 11 the SZ group), and 20 were complete (11 of the NSZ group and 9 of the SZ group). Of the final sample 40 WK, 16 were complete (11 of the NSZ group and 5 of the SZ group), and 24 were missing the interviewers' remarks (9 of the NSZ group and 15 of the SZ group).

### Narrative measures

**Nominal domain.** Subject pronouns were manually annotated considering their number and person composition and divided into two main classes based on their phonological form: null (N) and overt (O). Null 3Person subject pronouns were coded as either referential (+R), and non-referential (-R). Anomalies in the referential use of 3Person pronouns were classified as anomalous (+A), or non-anomalous (–A).

**Sentential domain.** All sentences were manually annotated and classified into matrix (MS) and embedded (ES). Syntactically incomplete matrix and embedded sentences were further marked as truncated anomalous (TS+A) or non-anomalous (TS–A).

**Linguistic variables.** We quantified five linguistic properties, three at the nominal domain and two at the sentential domain, manipulating twelve linguistic variables (**Table 3**). To control for the overall quantities of speech produced by each participant, all linguistic variables were calculated by dividing the total number of a target item (e.g., null pronouns) per number of words in the narrative and multiplied by 100.

**Statistical analysis.** All statistical analyses were conducted on the Statistical Package for the Social Sciences (SPSS) version 27 for MacBook. Data from samples 1 and 2 were analyzed separately in three stages: stage 1: linguistic variables (**Table 3**) were calculated for DM and WK separately, and significant group differences were determined; stage 2: correlation analyses between the linguistic variables with significant group effect obtained in stage 1 and the scores of the psychometric evaluations were conducted; stage 3: significant narrative differences were determined within each group. Most of our data were not normally distributed, thus, Mann–Whitney $U$ test comparisons were conducted to determine whether there were significant group and narrative differences. All indicated p-values are at the level of one-tail. Bonferroni adjustment to p-level was applied considering the number of variables analyzed, avoiding possible familywise error rate [64, 65]. Thus, the p-level of 3Person pronoun referential anomaly was adjusted to 0.013, according to the number of variables involved in the comparison. By contrast, the p-level of all other comparisons was set at 0.05. Eta squared ($\eta^2$) was calculated (based on Z values obtained for Mann Whitney 2 independent samples test) and used to quantify effect sizes whenever group differences reached significance at the level of 0.05. In accordance with Cohen's suggestions about interpretation of effect size magnitude, $\eta^2$ values are considered small ($\eta^2 < 0.01$), medium ($0.01 < \eta^2 < 0.25$), and large ($\eta^2 > 0.25$) [66]. Spearman's rho correlation coefficient was used to calculate correlations between linguistic variables with significant group differences at the p-level of 0.05 (2-tailed) and the scores of psychometric scales. After Bonferroni adjustment, the p-level of significance for correlations was set at 0.01.

**Table 3. Linguistic properties and their corresponding variables at sentential and nominal domains.**

| Nominal domain | Sentential domain |
|---|---|
| *Subject Pronoun phonological form* | *Sentence Type* |
| Overt Pronoun (O) | Matrix Sentences (MS) |
| Null Pronoun (N) | Embedded Sentences (ES) |
| *Null 3Person pronoun referential status* | *Truncation* |
| Null 3P Referential (N3P+R) | Truncated Anomalous (TS+A) |
| Null 3P Non-Referential (N3P-R) | Truncated Non-anomalous (TS-A) |
| *3Person referential pronoun anomaly status* | |
| Null 3P Referential Anomalous (N3P+R+A) | |
| Null 3P Referential Non-anomalous (N3P+R-A) | |
| Overt 3P Referential Anomalous (O3P+R+A) | |
| Overt 3P Referential Non-anomalous (O3P+R-A) | |

## Results

### Results from Sample 1

**Stage 1: Between-group analyses of DM and of WK narratives.** The SZ group produced a significantly smaller number of words in DM only (DM: $p < 0.001$; WK: $p = 0.640$). DM showed significant differences on null pronouns, matrix sentences, and truncated non-anomalous sentences, with a medium effect size (Table 4).

In WK, there were significant differences on null 3Person referential pronouns and null 3Person referentially non-anomalous pronouns, with medium effect size (Table 5). However, null 3Person referentially non-anomalous pronouns were not significant after the Bonferroni adjustment.

**Stage 2: Correlation analysis.** Significant correlations were found between the linguistic variables with significant group differences in DM and WK and the scores of the psychometric scales. Matrix sentences significantly correlated with most psychometric measures, and null pronouns with the PANSS positive subscale (Table 6).

**Stage 3: Between-narrative analyses within group.** Both groups produced more words when reporting a dream; however, only within the NSZ group was the difference between narratives significant (NSZ participants: $p < 0.001$, and participants with SZ: $p = 0.398$).

There was no significant narrative difference within the SZ group (Table 7); however, within the NSZ group, we found significant differences at the pronominal and sentential level.

We found significant narrative differences within the NSZ group (Table 8). There was a significantly lower proportion of null pronouns in general, and a significantly higher proportion of null 3Person referential pronouns, of null 3Person non-referential pronouns, and of null 3Person referentially non-anomalous pronouns in DM. At the sentence level, the NSZ group produced a higher proportion of embedded sentences and fewer matrix sentences in DM.

**Table 4. Between-group differences in dream narratives in Sample 1 (n = 40).**

| *Nominal Domain* | Mean (SD) | | Mann-Whitney U | Z | P | Effect Size |
|---|---|---|---|---|---|---|
| | SZ | NSZ | | | | |
| Pronoun phonological form | | | | | | |
| **Null Pronoun** | **10.15 (5.3)** | **7.28 (2.4)** | **115.500** | **-2.286** | **0.021** | **0.13** |
| Overt Pronoun | 7.20 (2.8) | 7.9 (2.1) | 160.000 | -1.082 | 0.289 | |
| Null 3P referential status | | | | | | |
| Null 3P Referential | 2.65 (2.2) | 2.04 (1.2) | 183.000 | -0.461 | 0.659 | |
| Null 3P Non-referential | 2.16 (2.2) | 2.03 (1.3) | 189.000 | -0.298 | 0.779 | |
| 3Person referential anomaly | | | | | | |
| Null Anomalous | 0.90 (1.3) | 0.53 (0.6) | 189.000 | -0.315 | 0.779 | |
| Null Non-anomalous | 1.75 (2.2) | 1.52 (1.0) | 170.500 | -0.801 | 0.429 | |
| Overt Anomalous | 0.39 (0.6) | 0.40 (1.3) | 170.000 | -1.031 | 0.429 | |
| Overt Non-anomalous | 0.87 (1.4) | 0.95 (0.9) | 156.500 | -1.225 | 0.242 | |
| *Sentential Domain* | Mean (SD) | | Mann-Whitney U | Z | *p* | Effect Size |
| | SZ | NSZ | | | | |
| Sentence Type | | | | | | |
| **Matrix Sentences** | **16.65 (2.9)** | **13.74 (2.3)** | **84.000** | **-3.138** | **0.001** | **0.25** |
| Embedded Sentences | 6.37 (3.1) | 6.81 (2.0) | 178.000 | -0.595 | 0.565 | |
| Truncation | | | | | | |
| Anomalous | 1.43 (1.3) | 0.94 (0.9) | 160.000 | -1.086 | 0.289 | |
| **Non-anomalous** | **0.06 (0.2)** | **0.38 (0.4)** | **109.000** | **-2.958** | **0.013** | **0.22** |

Mann-Whitney *U* comparisons between-group in production in dream narratives. Variables with significant differences are in bold.

**Table 5. Between-group differences in waking narratives in Sample 1 (n = 40).**

| Nominal Domain | Mean (SD) | | Mann-Whitney U | Z | P | Effect Size |
|---|---|---|---|---|---|---|
| | SZ | NSZ | | | | |
| Pronoun phonological form | | | | | | |
| Null Pronoun | 12.26 (6.4) | 11.05 (7.2) | 167.000 | -0.893 | 0.383 | |
| Overt Pronoun | 5.71 (3.4) | 5.67 (3.8) | 193.000 | -0.190 | 0.862 | |
| Null 3P referential status | | | | | | |
| **Null 3P Referential** | **3.24 (3.8)** | **0.84 (1.3)** | **101.500** | **-2.711** | **0.007** | **0.19** |
| Null 3P Non-referential | 1.22 (1.4) | 0.71 (0.9) | 163.500 | -1.036 | 0.327 | |
| 3Person referential anomaly | | | | | | |
| Null Anomalous | 0.87 (1.6) | 0.24 (0.4) | 177.000 | -0.748 | 0.547 | |
| Null Non-anomalous | 2.37 (3.2) | 0.60 (1.0) | 117.500 | -2.307 | **0.024** | 0.14 |
| Overt Anomalous | 0.38 (0.8) | 0.34 (0.9) | 183.500 | -0.639 | 0.659 | |
| Overt Non-anomalous | 0.73 (1.2) | 0.68 (0.9) | 200.000 | 0.000 | 1.000 | |
| *Sentential Domain* | Mean (SD) | | Mann-Whitney U | Z | p | Effect Size |
| | SZ | NSZ | | | | |
| Sentence Type | | | | | | |
| Matrix Sentences | 18.21 (4.7) | 16.84 (6.2) | 150.500 | -1.339 | 0.183 | |
| Embedded Sentences | 5.08 (3.4) | 4.72 (3.2) | 193.000 | -0.189 | 0.862 | |
| Truncation | | | | | | |
| Anomalous | 1.03 (1.3) | 1.06 (1.5) | 200.000 | 0.000 | 1.000 | |
| Non-anomalous | 0.36 (0.7) | 0.26 (0.6) | 183.000 | -0.605 | 0.659 | |

Mann-Whitney *U* comparisons between-group in production in waking narratives. Variables with significant differences are in bold. Note that the production of null 3Person referentially non-anomalous pronouns were not significant after Bonferroni adjustment.

## Results from Sample 2

**Stage 1: Between-group analyses of DM and of WK narratives.** Overall, the narratives of the SZ group showed significantly fewer words compared to the narratives produced by the NSZ group (DM: *p* = 0.003, and WK: *p* = 0.016).

In DM narratives, no significant differences of the linguistic variables analyzed were found; however, there was a significant difference of null 3Person referential pronouns in WK narratives (**Tables 9 and 10**).

**Table 6. Spearman's rho correlations—Sample 1 (n = 40).**

| | | BPRS | Total PANSS | Positive PANSS | Negative PANSS | General PANSS |
|---|---|---|---|---|---|---|
| *Matrix sentence* | Cor. Coef. | 0.389* | **0.447**** | **0.519**** | 0.378* | 0.306 |
| | Sig. (2-tailed) | 0.013 | **0.004** | **0.000** | 0.016 | 0.055 |
| *Truncated non-anomalous* | Cor. Coef. | -0.202 | -0.310 | -0.399* | **-0.407**** | -0.164 |
| | Sig. (2-tailed) | 0.210 | 0.052 | 0.011 | **0.009** | 0.313 |
| *Null pronouns* | Cor. Coef. | 0.350* | 0.360* | **0.481**** | 0.348* | 0.237 |
| | Sig. (2-tailed) | 0.027 | 0.023 | **0.002** | 0.028 | 0.141 |
| *Null 3Person referential* | Cor. Coef. | 0.366* | **0.420**** | **0.469**** | 0.367* | 0.332* |
| | Sig. (2-tailed) | 0.020 | **0.007** | **0.002** | 0.020 | 0.036 |

Spearman's rho correlations of the significant linguistic variables with the scores of BPRS, PANSS Total, PANSS Positive, PANSS Negative, and PANSS General. Significant correlations are in bold.

**Table 7. Within-group differences in production of participants with SZ in Sample 1 (n = 40).**

| Nominal Domain | Mean (SD) | | Mann-Whitney U | Z | P | Effect Size |
|---|---|---|---|---|---|---|
| | DM | WK | | | | |
| Pronoun phonological form | | | | | | |
| Null Pronoun | 10.15 (5.2) | 12.26 (6.4) | 161.500 | -1.041 | 0.301 | |
| Overt Pronoun | 7.20 (2.8) | 5.71 (3.4) | 160.500 | -1.069 | 0.289 | |
| Null 3P referential status | | | | | | |
| Null 3P Referential | 2.65 (2.2) | 3.24 (3.8) | 199.500 | -0.014 | 0.989 | |
| Null 3P Non-referential | 2.16 (2.2) | 1.22 (1.4) | 142.000 | -1.586 | 0.121 | |
| 3Person referential anomaly | | | | | | |
| Null Anomalous | 0.90 (1.3) | 0.87 (1.6) | 183.500 | -0.504 | 0.659 | |
| Null Non-anomalous | 1.75 (2.2) | 2.37 (3.2) | 186.000 | -0.384 | 0.718 | |
| Overt Anomalous | 0.39 (0.6) | 0.38 (0.8) | 183.500 | -0.551 | 0.659 | |
| Overt Non-anomalous | 0.87 (1.4) | 0.73 (1.2) | 195.500 | -0.135 | 0.904 | |
| *Sentential Domain* | Mean (SD) | | Mann-Whitney U | Z | *p* | Effect Size |
| | DM | WK | | | | |
| Sentence Type | | | | | | |
| Matrix Sentences | 16.65 (2.9) | 18.21 (4.7) | 154.500 | -1.231 | 0.221 | |
| Embedded Sentences | 6.37 (3.10) | 5.08 (3.4) | 142.000 | -1.569 | 0.121 | |
| Truncation | | | | | | |
| Anomalous | 1.43 (1.3) | 1.03 (1.3) | 156.000 | -1.223 | 0.242 | |
| Non-anomalous | 0.06 (0.2) | .36 (0.7) | 157.000 | -1.665 | 0.253 | |

Mann-Whitney *U* comparisons within-group in production of the SZ group.

**Table 8. Within-group differences in production of NSZ participants in Sample 1 (n = 40).**

| Nominal Domain | Mean (SD) | | Mann-Whitney U | Z | P | Effect Size |
|---|---|---|---|---|---|---|
| | DM | WK | | | | |
| Pronoun phonological form | | | | | | |
| **Null Pronoun** | **7.28 (2.4)** | **11.05 (7.2)** | **111.500** | **-2.394** | **0.015** | **0.15** |
| Overt Pronoun | 7.90 (2.1) | 5.67 (3.8) | 133.000 | -1.813 | 0.072 | |
| Null 3P referential status | | | | | | |
| **Null 3P Referential** | **2.04 (1.2)** | **.84 (1.3)** | **76.000** | **-3.390** | **0.001** | **0.29** |
| **Null 3P Non-referential** | **2.03 (1.3)** | **0.71 (0.9)** | **74.500** | **-3.431** | **0.000** | **0.30** |
| 3Person referential anomaly | | | | | | |
| Null Anomalous | 0.53 (0.6) | 0.24 (0.4) | 141.000 | -1.748 | 0.114 | |
| Null Non-anomalous | **1.52 (1.0)** | **0.60 (1.0)** | **78.000** | **-3.345** | **0.001** | **0.29** |
| Overt Anomalous | 0.40 (1.3) | 0.34 (0.9) | 192.500 | -0.306 | 0.841 | |
| Overt Non-anomalous | 0.95 (0.9) | 0.68 (0.9) | 153.000 | -1.323 | 0.211 | |
| *Sentential Domain* | Mean (SD) | | Mann-Whitney U | Z | *p* | Effect Size |
| | DM | WK | | | | |
| Sentence Type | | | | | | |
| **Matrix Sentences** | **13.74 (2.3)** | **16.84 (6.2)** | **120.500** | **-2.151** | **0.030** | **0.12** |
| **Embedded Sentences** | **6.81 (2.0)** | **4.72 (3.2)** | **119.000** | **-2.191** | **0.028** | **0.12** |
| Truncation | | | | | | |
| Anomalous | 0.94 (0.9) | 1.06 (1.5) | 169.000 | -0.853 | 0.414 | |
| Non-anomalous | 0.38 (0.4) | 0.26 (0.6) | 178.500 | -0.582 | 0.565 | |

Mann-Whitney *U* comparisons within-group in production of the NSZ group. Variables with significant differences are in bold.

Table 9. Between-group differences in dream narratives in Sample 2 (n = 31).

| Nominal Domain | Mean (SD) | | Mann-Whitney U | Z | P | Effect Size |
|---|---|---|---|---|---|---|
| | SZ | NSZ | | | | |
| Pronoun phonological form | | | | | | |
| Null Pronoun | 7.85 (5.2) | 6.64 (3.6) | 90.500 | -0.805 | 0.427 | |
| Overt Pronoun | 7.39 (5.0) | 7.63 (3.5) | 105.500 | -0.186 | 0.855 | |
| Null 3P referential status | | | | | | |
| Null 3P Referential | 2.30 (2.6) | 1.72 (2.1) | 100.500 | -0.412 | 0.699 | |
| Null 3P Non-referential | 0.43 (1.0) | 1.50 (2.4) | 72.000 | -1.717 | 0.123 | |
| 3Person referential anomaly | | | | | | |
| Null Anomalous | 0.61 (2.0) | 0 | 100.000 | -1.348 | 0.699 | |
| Null Non-anomalous | 1.69 (2.2) | 1.72 (2.1) | 105.000 | -0.219 | 0.855 | |
| Overt Anomalous | 0 | 0.06 (0.3) | 104.500 | -0.742 | 0.823 | |
| Overt Non-anomalous | 0.38 (1.3) | 1.44 (2.0) | 68.000 | -2.027 | 0.087 | |
| *Sentential Domain* | Mean (SD) | | Mann-Whitney U | Z | p | Effect Size |
| | SZ | NSZ | | | | |
| Sentence Type | | | | | | |
| Matrix Sentences | 17.08 (5.7) | 16.20 (4.6) | 109.500 | -0.021 | 0.984 | |
| Embedded Sentences | 4.32 (3.2) | 6.00 (3.1) | 81.000 | -1.200 | 0.244 | |
| Truncation | | | | | | |
| Anomalous | 3.84 (9.3) | 0.56 (.9) | 101.500 | -0.410 | 0.730 | |
| Non-anomalous | 0.22 (0.7) | 0.50 (1.0) | 94.000 | -0.958 | 0.528 | |

Mann-Whitney *U* comparisons between-group in production in dream narratives.

**Stage 2: Correlation analysis.** Given that the NSZ group was not psychometrically evaluated, only the scores of the SZ group entered the correlation analysis. Also, only the linguistic variable assessing the production of null 3Person referential pronouns showed significant difference between groups, and in WK only; thus, only null 3Person referential pronouns entered the correlation analysis. No significant correlation was found (**Table 11**).

**Stage 3: Between-narrative analyses within group.** No significant narrative difference of number of words was found within either group (SZ: *p* = 1.000, and NSZ: *p* = 0.091), and no significant narrative difference was found within the SZ group (**Table 12**). However, we found significant narrative differences within the NSZ group.

As described in **Table 13**, at the pronominal level, there were significant narrative differences on null pronouns, null 3Person referential pronouns, null 3Person referentially non-anomalous pronouns, and overt 3Person referentially non-anomalous pronouns. However, after the Bonferroni adjustment, overt 3Person referentially non-anomalous pronouns was not significant. At the sentential level, we found significant narrative differences on matrix sentence and embedded sentence, with NSZ participants producing more embedded and fewer matrix sentences in DM compared to WK narratives, with a medium effect size, as reported on **Table 13**.

## Discussion

Our main findings, putting together both samples (**Table 14**) indicate that, compared to NSZ, participants with SZ used significantly more null pronouns, null 3Person referential pronouns and matrix sentences, and significantly fewer truncated non-anomalous sentences. The SZ group produced more null 3Person referentially anomalous pronouns in all narratives across

**Table 10. Between-group differences in waking narratives in Sample 2 (n = 31).**

| *Nominal Domain* | Mean (SD) | | Mann-Whitney U | Z | P | Effect Size |
|---|---|---|---|---|---|---|
| | SZ | NSZ | | | | |
| **Pronoun phonological form** | | | | | | |
| Null Pronoun | 11.22 (5.1) | 15.90 (11.2) | 83.500 | -1.095 | 0.279 | |
| Overt Pronoun | 5.32 (5.2) | 5.39 (3.9) | 99.000 | -0.456 | 0.670 | |
| **Null 3P referential status** | | | | | | |
| **Null 3P Referential** | **3.08 (4.2)** | **0.31 (1.0)** | **58.000** | **-2.791** | **0.032** | **0.26** |
| Null 3P Non-referential | 1.24 (1.8) | 0.82 (1.3) | 100.500 | -0.458 | 0.699 | |
| **3Person referential anomaly** | | | | | | |
| Null Anomalous | 1.44 (2.6) | 0.11 (0.5) | 84.000 | -1.842 | 0.298 | |
| Null Non-anomalous | 1.64 (2.5) | 0.20 (0.6) | 77.000 | -1.975 | 0.183 | |
| Overt Anomalous | 0 | 0 | 110.000 | 0.000 | 1.000 | |
| Overt Non-anomalous | 0 | 0.29 (1.0) | 99.000 | -1.066 | 0.670 | |
| *Sentential Domain* | Mean (SD) | | Mann-Whitney U | Z | p | Effect Size |
| | SZ | NSZ | | | | |
| **Sentence Type** | | | | | | |
| Matrix Sentences | 20.80 (7.0) | 21.55 (9.0) | 105.000 | -0.207 | 0.855 | |
| Embedded Sentences | 2.37 (3.2) | 2.89 (2.5) | 91.500 | -0.781 | 0.451 | |
| **Truncation** | | | | | | |
| Anomalous | 2.85 (6.3) | 0.57 (1.2) | 82.000 | -1.442 | 0.261 | |
| Non-anomalous | 0.26 (0.9) | 0.13 (0.6) | 105.000 | -0.485 | 0.855 | |

Mann-Whitney *U* comparisons between-group in production in waking narratives. Variables with significant differences are in bold.

samples, although no significant difference of 3Person referential anomalies was found. Results showed that both DM and WK were informative of SZ at the pronominal level; however, at the sentential level, only DM narratives were informative of SZ.

Our predictions were partially supported and partially unsupported. Although our analyses show that participants with SZ do tend to produce higher proportions of null pronouns, P.1 was only partially supported, since group differences did not reach significance across all comparisons; significant differences were only found in DM narratives of Sample 1. The overuse of null pronouns in general found in the narratives of both of our SZ groups across samples is in direct contrast with the ongoing reduction of null pronouns in CBP [57]. At first sight, however, the dialogic nature of our narrative samples could be considered a confounding variable, as it may have had an impact on the use of null pronouns. Along these lines, Simões discusses the high frequency of null subjects in narratives of children acquiring CBP [67]. Nonetheless, given that the narratives of the NSZ groups show a lower proportion of null pronouns compared to that of the SZ groups, observations about the nature of our data do not fully explain the constant higher proportion of null pronouns, particularly of null 3Person referential

**Table 11. Spearman's rho correlations (Sample 2 / n = 11).**

| | | BPRS | Total PANSS | Positive PANSS | Negative PANSS | General PANSS |
|---|---|---|---|---|---|---|
| *Null 3Person Referential* | Cor. Coef. | 0.085 | 0.220 | 0.295 | 0.339 | -0.144 |
| | Sig. (2-tailed) | 0.804 | 0.516 | 0.379 | 0.307 | 0.673 |

Spearman's rho correlations of the significant linguistic variable in waking narratives with the scores of BPRS, PANSS Total, PANSS Positive, PANSS Negative, and PANNS General of the SZ group.

**Table 12. Within-group differences in production of participants with SZ in Sample 2 (n = 22).**

| Nominal Domain | Mean (SD) | | Mann-Whitney U | Z | P | Effect Size |
|---|---|---|---|---|---|---|
| | DM | WK | | | | |
| Pronoun phonological form | | | | | | |
| Null Pronoun | 7.85 (5.2) | 11.22 (5.1) | 36.000 | -1.612 | 0.116 | |
| Overt Pronoun | 7.39 (5.0) | 5.32 (5.2) | 44.500 | -1.057 | 0.300 | |
| Null 3P referential status | | | | | | |
| Null 3P Referential | 2.30 (2.6) | 3.08 (4.2) | 55.500 | -0.345 | 0.748 | |
| Null 3P Non-referential | 0.43 (1.0) | 1.24 (1.8) | 47.500 | -1.088 | 0.401 | |
| 3Person referential anomaly | | | | | | |
| Null Anomalous | 0.61 (2.0) | 1.44 (2.6) | 50.000 | -1.025 | 0.519 | |
| Null Non-anomalous | 1.69 (2.2) | 1.64 (2.5) | 59.000 | -.111 | 0.949 | |
| Overt Anomalous | 0 | 0 | 60.500 | 0.000 | 1.000 | |
| Overt Non-anomalous | 0.38 (1.3) | 0 | 55.000 | -1.000 | 0.748 | |
| *Sentential Domain* | Mean (SD) | | Mann-Whitney U | Z | *p* | Effect Size |
| | DM | WK | | | | |
| Sentence Type | | | | | | |
| Matrix Sentences | 17.08 (5.7) | 20.80 (7.0) | 36.000 | -1.612 | 0.116 | |
| Embedded Sentences | 4.32 (3.2) | 2.37 (3.2) | 39.000 | -1.462 | 0.171 | |
| Truncation | | | | | | |
| Anomalous | 3.84 (9.3) | 2.85 (6.3) | 58.000 | -0.184 | 0.898 | |
| Non-anomalous | 0.22 (0.7) | 0.26 (0.9) | 60.000 | -0.066 | 1.000 | |

Mann-Whitney *U* comparisons within-group in production of the SZ group.

**Table 13. Within-group differences in production of NSZ participants in Sample 2 (n = 40).**

| Nominal Domain | Mean (SD) | | Mann-Whitney U | Z | P | Effect Size |
|---|---|---|---|---|---|---|
| | DM | WK | | | | |
| Pronoun phonological form | | | | | | |
| **Null Pronoun** | **6.64 (3.6)** | **15.90 (11.2)** | **85.500** | **-3.098** | **0.001** | **0.25** |
| Overt Pronoun | 7.63 (3.5) | 5.39 (3.9) | 129.000 | -1.922 | 0.056 | |
| Null 3P referential status | | | | | | |
| **Null 3P Referential** | **1.72 (2.1)** | **0.31 (1.0)** | **111.000** | **-2.893** | **0.015** | **0.21** |
| Null 3P Non-referential | 1.50 (2.4) | 0.82 (1.3) | 164.000 | -1.053 | 0.341 | |
| 3Person referential anomaly | | | | | | |
| Null Anomalous | 0 | 0.11 (.5) | 190.000 | -1.000 | 0.799 | |
| **Null Non-anomalous** | **1.72 (2.1)** | **0.20 (0.6)** | **106.000** | **-3.055** | **0.010** | **0.23** |
| Overt Anomalous | 0.06 (.3) | 0 | 190.000 | -1.000 | 0.799 | |
| Overt Non-anomalous | 1.44 (2.0) | 0.29 (1.0) | 122.000 | -2.603 | 0.035 | 0.17 |
| *Sentential Domain* | Mean (SD) | | Mann-Whitney U | Z | *p* | Effect Size |
| | DM | WK | | | | |
| Sentence Type | | | | | | |
| **Matrix Sentences** | **16.20 (4.6)** | **21.55 (9.0)** | **116.500** | **-2.259** | **0.023** | **0.13** |
| **Embedded Sentences** | **6.00 (3.1)** | **2.89 (2.5)** | **85.000** | **-3.119** | **0.001** | **0.25** |
| Truncation | | | | | | |
| Anomalous | 0.56 (0.9) | 0.57 (1.2) | 180.500 | -0.670 | 0.602 | |
| Non-anomalous | 0.50 (1.0) | 0.13 (0.6) | 161.500 | -1.676 | 0.301 | |

Mann-Whitney *U* comparisons within-group in production of the NSZ group. Variables with significant differences are in bold.

**Table 14. Performance of participants with SZ versus NSZ participants across narratives and samples.**

|  | Sample 1 | | Sample 2 | |
| --- | --- | --- | --- | --- |
|  | **DM** | **WK** | **DM** | **WK** |
| *Nominal Domain* | | | | |
| Null Pronouns | +* | (n.s) | (n.s) | (n.s) |
| Overt Pronouns | (n.s) | (n.s) | (n.s) | (n.s) |
| **Null 3Person Referential** | **(n.s)** | +* | **(n.s)** | +* |
| Null 3Person Non-Referential | (n.s) | (n.s) | (n.s) | (n.s) |
| **Null 3Person Referentially Anomalous** | **(n.s)** | **(n.s)** | **(n.s)** | **(n.s)** |
| Null 3Person Referentially Non-Anomalous | (n.s) | (n.s) | (n.s) | (n.s) |
| Overt 3Person Referentially Anomalous | (n.s) | (n.s) | (n.s) | (n.s) |
| Overt 3Person Referentially Non-Anomalous | (n.s) | (n.s) | (n.s) | (n.s) |
| *Sentential Domain* | | | | |
| Matrix Sentence | +* | (n.s) | (n.s) | (n.s) |
| Embedded Sentence | (n.s) | (n.s) | (n.s) | (n.s) |
| Truncated Anomalous | (n.s) | (n.s) | (n.s) | (n.s) |
| Truncated Non-Anomalous | (−)* | (n.s) | (n.s) | (n.s) |

Performance of the SZ groups in all linguistic variables, where "+" indicates higher means and "(−)" indicate lower means compared to the NSZ group.

(*) indicates significant group differences, and (n.s.) indicates non-significant group differences. Constant trends across all narrative types and studies are in bold.

pronouns in the SZ group, which reached significant level in the WK narratives of both samples. Hence, while acknowledging this possible confounding factor, we emphasize that the SZ groups used null referential 3Person pronouns in a rather deviant way.

Theoretical studies on CBP indicate that the frequency of null 3Person referential pronouns is dropping both in adult speakers and in children acquiring CBP. The literature shows that 39% of all 3Person referential pronouns found in a sample of narratives collected in 1995 were null, while in a sample collected in 2009/10, 28% were null pronouns [57]. As for children acquiring CBP, Lopes reported that 32% of all pronominal subjects produced were null [59], which, compared to the 55,5% frequency reported in Simões [67], shows a significant decrease. The frequency of null 3Person referential pronouns found in our samples of narratives of patients with SZ (sample 1 = 66%; sample 2 = 86%) is far higher than that reported in the literature for both typical adults and children speakers of CBP. The frequency of null 3Person referential pronouns found in our sample of narratives of NSZ group (sample 1 = 56%; sample 2 = 58%) is not only close to the frequency reported by the literature, but quite similar to the frequency reported in Simões [67]. Hence, the higher proportion of null 3Person referential pronouns shows that the use of null subject pronouns by patients with SZ is not in accordance with what is observed in CBP. Furthermore, according to the literature in partial null subject languages (including CBP), null 3Person referential pronouns are more frequent in embedded contexts taking the closest nominal expression as their antecedent [52, 55, 58, 68]. Thus, given the higher rate of matrix independent sentences and the low rate of embedded sentences, most of the null 3Person referential pronouns found in our corpora of SZ narratives arguably occurred in the context of independent matrix clauses, which reinforces the idea that the SZ groups used null pronouns in a deviant way. This observation, in addition to the fact that these speakers produced a high proportion of null 3Person subject pronouns, suggests a negligence of the structural and semantic constraints on null subject pronouns in SZ.

The significant correlation found between null pronouns and PANSS positive scores suggests that the overuse of null pronominal forms is related to positive symptoms of SZ. Whereas the significant correlations of null 3Person referential pronouns with the scores of Total PANSS and Positive PANSS suggest that the overuse of null 3Person referential pronouns is associated with symptoms of SZ in general, but more so with positive symptoms. However, let us observe that both null pronouns and null 3Person referential pronouns were less significantly correlated with FTD, thought disorder, as measured by PANSS P2 scores ([N]: $r = 0.419$; $p = 0.007$, and [N3P+R]: $r = 0.402$; $p = 0.010$). Hence, in tension with previous observations that language issues pattern together with FTD in SZ [30, 51], our results indicate that atypical overuse of null pronouns is significantly associated with positive symptoms in general, but less so with FTD.

Our second prediction (P.2) was not supported by our results: No significant group difference in terms of referential anomaly was found. But, again, the dialogic nature of the narratives might have influenced the lack of significant differences, since pronominal reference might have been restricted by the interviewer prompting questions and interventions. Most studies reporting referential anomalies [23, 29, 30 among others] adopt tasks in which the referential process is constrained within a given background, in which participants might be asked, for example, to recount a fairytale [29] or tell a story based on comic strips [30]. Also, to avoid false positives, whenever we were unsure, or there were disagreements among us, about a pronoun being anomalous, we did not annotate it as anomalous. Therefore, we conclude that, due to methodological issues, our samples are not informative about pronominal anomaly in SZ, but it is important to notice that the SZ groups produced significantly more null 3Person referential pronouns in WK narratives, and more null 3Person referentially anomalous pronouns in all narrative types across samples. Tovar et al.—as far as we know, the only study that investigated the use of null pronouns in patients with SZ—reported more errors in the use of null referential pronouns as compared to overt pronouns [51]. Our finding converges with that of Tovar et al.'s with respect to that. A closer look into our samples of narratives produced by patients with SZ reveals more anomalous null 3Person pronouns compared to overt ones, although overt/null differences in terms of referential anomaly did not reach significance in either sample 1 ($p = 0.055$) or sample 2 ($p = 0.080$). However, since our narratives elicited a greater number of null pronouns and few overt 3Person pronouns (in all comparisons), we found significant higher proportions of null compared to overt 3Person pronouns (sample 1: SZ and NSZ $p < 0.001$; and sample 2: SZ $p = 0.012$, and NSZ $p < 0.004$), we must consider two things. First, the tendency towards more null 3Person referentially anomalous pronouns can possibly characterize a ceiling effect. Second, the absence of significant differences in terms of anomalous use of overt 3Person pronouns might be because our narratives elicited very few overt 3Person pronouns.

Importantly, despite the lack of significant group difference, the consistent tendency of more null 3Person referentially anomalous pronouns observed in our samples of SZ narratives across experiments is in contrast with findings of first and second language acquisition studies. Typical children's knowledge of weak pronouns, which includes null and clitic forms, match that of adults [69–72]. Similarly, second language speakers do not differ from monolinguals with respect to the use of null pronouns [73, 74]. In comparison, our study shows that speakers with SZ do not behave linguistically as children and second language speakers.

Our predictions P.3 and P.4 were only partially supported. At sentence level, only DM narratives of sample 1 showed significant group differences, with the SZ group producing significantly more matrix sentences and fewer truncated non-anomalous sentences. Nevertheless, the tendency of producing more matrix and fewer truncated non-anomalous sentences was observed across narratives of both samples.

These findings are in line with studies indicating reduction in syntactic complexity in SZ [30–34, 37, 75, 76], being thus aligned with Hinzen and Rossello's hypothesis that SZ results in

an inability to build complex grammatical structures and complex relations between propositions [15].

Furthermore, the association between the overuse of simple sentences and SZ symptoms in general, as indicated by the significant correlation of matrix sentences with the Total PANSS scores, reinforces the general hypothesis of SZ causing reduction of syntactic complexity, while the significant correlation with the Positive PANSS scores supports Hinzen and Rossello's hypothesis that the impairment in the ability to build complex sentences and propositions are distinctive of SZ patients with positive symptoms (e.g., FTD and Delusions). As for the significant anticorrelation found between truncated non-anomalous sentences and Negative PANSS scores, it suggests that the ability to properly elide grammatical elements seems to be affected by the presence of negative symptoms.

Turning to the analysis of narrative type, recall that the graph analysis, using basic structural features from speech graphs, showed that only DM were informative of psychosis [28, 60], while our in-depth grammatical analysis showed both DM and WK narratives to be informative. WK showed significant between-group differences, while DM showed different results across samples. Thus, one of our driving questions was why one corpus of DM but not the other would show significant differences. Potential confounds could arise from the differences in the protocols adopted for the interviews that generated samples 1 and 2: narrative length (no limitation vs. 30 sec time-limit); participant's age (adults vs. teenagers); and number of psychotic episodes (multiple vs. one).

According to previous studies [6, 77–81], speech samples shorter than 10-minutes do not provide sufficient linguistic material for analyses of structural and referential failures. Tovar et al., who also examined interview narratives, collected 5-page reports, resulting in a mean number of 888.6 words [51]. The mean number of words in DM was 207.8 ± 52.7 in sample 1, and 53.2 ± 26.2 in sample 2. Also, when the proportion of sentences per number of words produced in DM is considered, in sample 1, the SZ group produced a higher proportion of sentences compared to the production of the NSZ group (SZ: mean of 23.02 ± 3.5, and NSZ: mean of 20.55 ± 1.9). While, in sample 2, the SZ group produced a smaller proportion of sentences (SZ: mean of 21.40 ± 6.0, and NSZ: mean of 22.20 ± 4.7). Therefore, narrative length can be a confounding variable for DM only in sample 2.

Moving to the possibility that the lack of significant group differences in DM of sample 2 was due to the participants' respective ages; this association is rather unlikely. Language acquisition studies show that by the age of 3 years old, children are just like adults in their ability to produce and understand sentences, to judge the truth value of sentences, and to understand the relations between sentences [82, 83]. In fact, according to the continuity hypothesis, typically-developing children internalize the grammatical system of their mother tongue before the age at which they start receiving formal education [84]. All participants of the sample 2 were teenagers (SZ: age mean of 14.64 ± 2.6; NSZ: age mean of 15.80 ± 3.3). In addition, between-narrative analyses within both the SZ and NSZ groups show the same trend across experiments. That is, we observed no significant narrative differences within the SZ groups, and pretty much the same differences within the NSZ groups. It is to be acknowledged, nevertheless, that possible delays in language acquisition in at-risk populations might be an early prodrome symptom of the disorder [10, 85, 86], underlying, thus, drastic deterioration of linguistic abilities as the individual gets older [34]. Hence, although it is unlikely that the lack of differences at the significant level observed in sample 2 is related to the participants' younger age, differences found in DM of sample 1 might be due to a putative deterioration of grammar related to the chronicity of the disorder, and possibly to the long-term effect of antipsychotic medication [87]. Either way, patients with multiple psychotic episodes might exhibit more severe grammatical deficits.

Participants with SZ showed a reduced ability to generate grammatical truncated sentences, suggesting anomalies at the sentential domain. This observation reached significance in sample 1, but not in sample 2, reinforcing, thus, the conclusion that the chronicity of the disorder plays a role in language disintegration.

There are three main limitations to our investigation. First, the different protocols adopted across experiments, which might have affected the analysis of DM. Second, the lack of detailed information regarding the clinical characteristics and the dosage of antipsychotic medication of the patients with SZ, particularly subjects of sample 1, prevented the detailed characterization of the group with SZ according to specific diagnoses; thus, no distinction between specific language symptoms (e.g., thought disorder versus non-disorder) could be drawn from our study. Third, participants were not evaluated in terms of working memory, executive function, or Theory of Mind (ToM); thus, our data cannot be interpreted in terms of these other cognitive abilities.

Overall, the present study contributes to the research on SZ-specific linguistic profile and highlights the importance of crosslinguistic data and formal analysis in increasing our understanding of syndrome-specific language features. We offer evidence from a partial pro-drop language that SZ affects the grammar of pronouns. The present investigation is the first of its nature focusing on the grammar of patients with SZ native speakers of Portuguese. Our results strongly suggest difficulties related to the licensing of null pronouns in SZ, and they point to the fact that the knowledge of pronouns in patients with SZ differs from that of typically developing children acquiring first language and from that of adults acquiring second language. Our results also indicate that the chronicity of the disorder might affect the grammar of SZ, compromising it even further. At this point, we highlight the need for further investigations on the effect of antipsychotic medication on language dysfunctions, as current findings show evidence of improvement in the language scores of the Repeatable Battery for the Assessment of Neuropsychological Status (RBANS) under the effect of amisulpride augmentation therapy [88]. In addition, our results show the importance of acknowledging the potential role of the narrative tasks in eliciting the linguistic material for the intended analysis.

Beyond these limitations, and in line with previous results [23–30], our study points to deficiencies at the pronominal and sentential levels as a cross-cultural linguistic marker of SZ.

## Supporting information

**S1 Appendix. Annotation scheme and narrative examples of samples 1 and 2.**
(DOCX)

**S2 Appendix. Definition of the analyzed language variables, with examples.**
(DOCX)

**S3 Appendix. Analyzed variables and final data of dream and waking narratives of samples 1 and 2.**
(DOCX)

**S4 Appendix. Full regression data.**
(DOCX)

## Acknowledgments

We want to thank Juan Uriagereka, Wolfram Hinzen and Rogério Panizzutti for the insightful comments and discussions, and the blind raters Cristina Prim and Marco Carreira for their amazing contribution to our research.

## Author Contributions

**Conceptualization:** Monica F. Chaves, Cilene Rodrigues, Sidarta Ribeiro.

**Data curation:** Monica F. Chaves, Cilene Rodrigues, Natália B. Mota, Mauro Copelli.

**Formal analysis:** Monica F. Chaves, Cilene Rodrigues.

**Funding acquisition:** Monica F. Chaves, Cilene Rodrigues.

**Investigation:** Monica F. Chaves.

**Methodology:** Monica F. Chaves, Cilene Rodrigues, Sidarta Ribeiro, Natália B. Mota, Mauro Copelli.

**Project administration:** Cilene Rodrigues, Sidarta Ribeiro.

**Supervision:** Cilene Rodrigues, Sidarta Ribeiro.

**Writing – original draft:** Monica F. Chaves, Cilene Rodrigues, Sidarta Ribeiro.

**Writing – review & editing:** Monica F. Chaves, Cilene Rodrigues, Sidarta Ribeiro.

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
