## [Decision Letter · Decision Letter 0]

29 Jun 2023

PONE-D-23-07587Grammatical impairment in schizophrenia: an exploratory study of the pronominal and sentential domainsPLOS ONE

Dear Dr. de Freitas Frias Chaves,

Thank you for submitting your manuscript to PLOS ONE. After careful consideration, we feel that it has merit but does not fully meet PLOS ONE’s publication criteria as it currently stands. Therefore, we invite you to submit a revised version of the manuscript that addresses the points raised during the review process.

We look forward to receiving your revised manuscript.

Kind regards,

Michal Ptaszynski, PhD

Academic Editor

PLOS ONE

Journal Requirements:

Reviewers' comments:

Reviewer's Responses to Questions

**Comments to the Author**

1. Is the manuscript technically sound, and do the data support the conclusions?

Reviewer #1: Partly

Reviewer #2: Yes

2. Has the statistical analysis been performed appropriately and rigorously? 

Reviewer #1: Yes

Reviewer #2: Yes

3. Have the authors made all data underlying the findings in their manuscript fully available?

Reviewer #1: Yes

Reviewer #2: Yes

4. Is the manuscript presented in an intelligible fashion and written in standard English?

Reviewer #1: Yes

Reviewer #2: Yes

5. Review Comments to the Author

Reviewer #1: In the present study, the authors compared the way individuals diagnosed with schizophrenia, who are native speakers of Colloquial Brazilian Portuguese (CBP), operationalize language with the way they speak in a control group, and identified several differences.

Schizophrenia is a psychiatric disorder characterized by thought disorders and impaired associations between semantic concepts. The study revealed that there are notable differences in the SZ group regarding the operationalization of words, occurring slightly before the establishment of meaning. The paper has been meticulously written, although I believe the discussion section could be more concise. It is evident that this study faced significant challenges, particularly in terms of research methodology.

However, one aspect that this study could further elaborate on is the nature of language and thought disorders in schizophrenia, as mentioned by the author from the beginning (starting from line 55). The results of the present study seem to primarily describe qualitative differences among speakers of a specific linguistic region. It is challenging to ascertain why schizophrenic patients with CBP develop such differences and how these differences correlate with their pathology.

To better understand the association with pathology, it would be beneficial to provide more detailed information about the characteristics of the patient group. Schizophrenia is known for its heterogeneity, and a more comprehensive description of the clinical characteristics of the schizophrenia group (n=20) in Sample 1 would be highly valuable. For instance, could you provide information regarding the dosage of antipsychotic medication? The variations in results between sample 1 and sample 2 appear to be linked to the long-term usage of antipsychotic medication, as discussed by the authors.

Regarding the correlation analysis, particularly Table 6, it seems necessary to conduct a multiple regression analysis while adjusting for age and educational history.

I noticed that Sample 1 includes psychosis as a control. To prevent confusion, it would be less ambiguous to consistently refer to the non-schizophrenic group rather than using "CT" in the text.

Considering the depth and significance of the findings, you might want to consider submitting the article to a specialized journal focused on schizophrenia. Doing so could attract even more attention from readers interested in this specific area of research.

Reviewer #2: This is an interesting study which found that deficits at the pronominal and sentential levels constitute a cross-cultural linguistic marker of SZ. There were several comments which should be addressed.

1. Schizophrenia is a severe psychiatric disorder characterized by positive symptoms, negative symptoms, and cognitive deficits. Schizophrenia (SZ) is associated with a variety of grammatical deficits. Grammatical deficits is one of the type of cognitive deficits. Recent study showed that amisulpride improved language performence compared with placebo in SZ (Zhu et al., Amisulpride augmentation therapy improves cognitive performance and psychopathology in clozapine-resistant treatment-refractory schizophrenia: a 12-week randomized, double-blind, placebo-controlled trial). I suggest the authors should noted it in the background or discussion section.

2. How to calculate the sample size? Can these sample sizes achieve statistical efficacy？

3. Please interpret why the sample is divided into two sample groups. What is the purpose of splitting the sample into two groups in this way? I suggest stating in the paper.

6. PLOS authors have the option to publish the peer review history of their article (what does this mean?). If published, this will include your full peer review and any attached files.

Reviewer #1: No

Reviewer #2: No

---

## [Author Response · Author response to Decision Letter 0]

13 Aug 2023

Dear Mr. Ptaszynski,

Thank you for your comments. I would like make clear that the minimal data set underlying the results described in the manuscript can be found within the supporting information files (S1 -S4 appendix files), as stated in Data Availability statement. I would, also, like to ensure you that there is no ethical or legal restrictions to sharing our data publicly.

The comments of both reviewers were dully addressed in the rebuttal letter.

In case you need more information, please, do not hesitate to ask.

Kind regards,

Monica de Freitas Frias Chaves

---

## [Decision Letter · Decision Letter 1]

30 Aug 2023

Grammatical impairment in schizophrenia: an exploratory study of the pronominal and sentential domains

PONE-D-23-07587R1

Dear Dr. de Freitas Frias Chaves,

We’re pleased to inform you that your manuscript has been judged scientifically suitable for publication and will be formally accepted for publication once it meets all outstanding technical requirements.

Kind regards,

Michal Ptaszynski, PhD

Academic Editor

PLOS ONE

Additional Editor Comments (optional):

Reviewers' comments:

Reviewer's Responses to Questions

**Comments to the Author**

1. If the authors have adequately addressed your comments raised in a previous round of review and you feel that this manuscript is now acceptable for publication, you may indicate that here to bypass the “Comments to the Author” section, enter your conflict of interest statement in the “Confidential to Editor” section, and submit your "Accept" recommendation.

Reviewer #1: All comments have been addressed

Reviewer #2: All comments have been addressed

2. Is the manuscript technically sound, and do the data support the conclusions?

Reviewer #1: Yes

Reviewer #2: Yes

3. Has the statistical analysis been performed appropriately and rigorously? 

Reviewer #1: Yes

Reviewer #2: Yes

4. Have the authors made all data underlying the findings in their manuscript fully available?

Reviewer #1: Yes

Reviewer #2: Yes

5. Is the manuscript presented in an intelligible fashion and written in standard English?

Reviewer #1: Yes

Reviewer #2: Yes

6. Review Comments to the Author

Reviewer #1: I was pleased to see your courteous response to my comments. Your thorough reply has left me thoroughly satisfied.

Reviewer #2: Thank you for your revision. All my concerns have been addressed. I think it should be published now.

7. PLOS authors have the option to publish the peer review history of their article (what does this mean?). If published, this will include your full peer review and any attached files.

Reviewer #1: No

Reviewer #2: **Yes: **ZEZHI LI

---

## [Editor Report · Acceptance letter]

4 Sep 2023

PONE-D-23-07587R1 

Grammatical impairment in schizophrenia: an exploratory study of the pronominal and sentential domains 

Dear Dr. Chaves:

I'm pleased to inform you that your manuscript has been deemed suitable for publication in PLOS ONE. Congratulations! Your manuscript is now with our production department. 

Kind regards, 

on behalf of

Dr. Michal Ptaszynski 

Academic Editor

PLOS ONE